# In Vivo Assessment of High-Strength and Corrosion-Controlled Magnesium-Based Bone Implants

**DOI:** 10.3390/bioengineering10070877

**Published:** 2023-07-24

**Authors:** Hamdy Ibrahim, Caroline Billings, Moataz Abdalla, Ahmed Korra, David Edger Anderson

**Affiliations:** 1Department of Mechanical Engineering, University of Tennessee, Chattanooga, TN 37403, USA; fjx788@mocs.utc.edu (M.A.); hwz711@mocs.utc.edu (A.K.); 2College of Veterinary Medicine, University of Tennessee, Knoxville, TN 37996, USA; cbilli10@vols.utk.edu (C.B.); dander48@utk.edu (D.E.A.)

**Keywords:** bone implants, biodegradable, in vivo, in vitro, magnesium

## Abstract

The biodegradable nature of magnesium in aqueous mediums makes it an attractive material for various biomedical applications when it is not recommended that the material stay permanently in the body. Some of the main challenges that hinder the use of magnesium for bone fracture repair are its limited mechanical strength and fast corrosion rates. To this end, we developed a novel Mg-Zn-Ca-Mn-based alloy and post-fabrication methods that can deliver high-strength and corrosion-controlled implant materials to address these challenges. This study is focused on assessing the in vitro corrosion and in vivo biocompatibility of the developed magnesium-based alloy and post-fabrication processes. The developed heat treatment process resulted in an increase in the microhardness from 71.9 ± 5.4 HV for the as-cast Mg alloy to as high as 98.1 ± 6.5 HV for the heat-treated Mg alloy, and the ceramic coating resulted in a significant reduction in the corrosion rate from 10.37 mm/yr for the uncoated alloy to 0.03 mm/yr after coating. The in vivo assessments showed positive levels of biocompatibility in terms of degradation rates and integration of the implants in a rabbit model. In the rabbit studies, the implants became integrated into the bone defect and showed minimal evidence of an immune response. The results of this study show that it is possible to produce biocompatible Mg-based implants with stronger and more corrosion-controlled properties based on the developed Mg-Zn-Ca-Mn-based alloy and post-fabrication methods.

## 1. Introduction

Bone fractures are a serious public health concern worldwide, with new fractures affecting an estimated 178 million individuals globally in 2019 [1]. Bone fractures negatively impact affected individuals, leading to decreased productivity, high healthcare costs, and impaired quality of life, including the risk of depressive thoughts [2,3]. For fractures that cannot be stabilized with simple casting or splinting alone, treatment is reliant upon metallic fixation devices [4] such as bone screws, plates, pins and wires, and reconstruction meshes [5,6,7,8,9]. Such devices facilitate the stabilization of complex fractures [10] and aid patients in regaining form and function in the affected limb or body region. The decision whether to remove fixation devices post fracture resolution or to leave them indwelling is often debated. Removal of the implant places a burden on the healthcare system and may be accompanied by complications [11,12]; however, hardware left indwelling poses risks of hypersensitivity reactions [7], infection [13], stress-shielding and weakening of the bone surrounding implants [12,14], pain, or impaired function [11].

The development of biocompatible, biodegradable materials to reduce reliance on long-term metallic implants is an area of interest [14] that holds the potential to reduce material-associated complications and reduce the overall burden on the healthcare system [15,16]. Mg is a lightweight metal that is naturally present in the human body, with its primary storage being located within bone [14]. Mg is biodegradable within aqueous media and is recognized to be appropriately biocompatible [17]. An additional main attraction of Mg is its perceived mechanical composition profile. Mg alloys are believed to possess mechanical characteristics, including a modulus of elasticity, that are more similar to those of bone when compared to stainless steel and titanium alloys, both common metals utilized in orthopedic implants [18]. These properties make Mg an attractive material for biomedical applications, including fracture repair.

Controlling the corrosion rate of Mg alloys and Mg-based metals is an area that requires attention. If not well-controlled, Mg degradation can generate a basic pH that impedes tissue healing or creates hydrogen gas byproducts [18]. The degradation of Mg-based implants requires fine-tuning to ensure that degradation occurs concomitantly with bone healing and also to avoid negative impacts on tissue healing. A numerical approach for the prediction of the biodegradation of Mg alloy implants was made by our team [19]. Alloying and post-fabrication processes (e.g., heat treatments, mechanical treatments, and coatings) have been investigated to produce biodegradable Mg-based implants with the high strength and slow corrosion rates needed to provide the required stability to the fractured bone during the healing period. For instance, Mg-Zn-Ca-based alloys have shown a great potential as biodegradable materials due to the high biocompatibility of the used alloying elements (i.e., Zn and Ca) and their improved mechanical and corrosion properties [15,20,21,22]. And the addition of a small amount of Mn (up to 0.5% wt.%) to Mg-Zn-Ca-based alloys has been shown to result in a significant grain size reduction, from 216 µm to 67 µm, and, hence, enhanced mechanical and corrosion properties [23,24]. In addition, heat treatment is one of the most effective methods for enhancing the mechanical characteristics and corrosion behavior of Mg alloys, as heat treatment merely affects the microstructure of an alloy without changing the chemical composition, geometry, or shape of the material [25].

Our previous experimental work demonstrated that a significant increase in mechanical strength and corrosion resistance can be achieved after performing a novel heat treatment process on various Mg-Zn-Ca-based alloy systems, compared to the pure Mg and as-cast alloys. The developed heat treatment process utilizes a solution treatment process followed by quenching and age-hardening to achieve sufficient strength and appropriate stiffness. Among the investigated alloy systems (e.g., Mg-1.2Zn-0.5Ca [26,27], Mg-1.2Zn-0.5Ca-0.5Mn [28], and Mg-1.6Zn-0.5Ca [29] alloys), the Mg-1.2Zn-0.5Ca-0.5Mn (wt.%) alloy showed the most promising mechanical and corrosion properties. This enhancement in mechanical and corrosion properties was attributed to the uniform distribution of the secondary intermetallic phases into fine nano precipitates in the microstructure of the alloy. Despite the enhancement in the corrosion behavior of the developed alloys after the heat treatments, coating them with biocompatible materials is still a necessity to further control the corrosion rates and enhance bone healing.

Several types of coatings have been investigated to improve the corrosion resistance and surface hardening of Mg alloys that can be classified into bioinert coatings and bioactive coatings. Among the different coating techniques used, micro arc oxidation (MAO) is one of the most successful processes to create a coating layer that is tightly bonded to the Mg alloy substrate with a porous surface morphology that allows for the deposition of subsequent coating layers as needed [24,26,30]. This can be beneficial when the inclusion of nanomaterials of different elements to the implant’s surface is required, such as hydroxyapatite, gadolinium, or iron/zinc [31,32].

To improve the surface hardness and corrosion resistance of Mg alloys, a micro arc oxidation (MAO) coating process is used on them to prepare ceramic coatings [30,33,34]. The main purpose of the MAO coating is to prevent the rapid corrosion rate during the healing period at the location of the implant. The creation of a MAO coating layer on Mg alloys is currently a well-developed and universal technique that can be used to increase the corrosion resistance of Mg alloys for bone implant applications [35]. Its use on different Mg-Zn-Ca-based alloy systems was presented in our previous work and in the literature [23,24,26,36]. In our latest work [24], the cytotoxicity of this novel material was investigated using a cytotoxicity dilution protocol and according to the ISO 10993-5 standard. The heat-treated and coated groups showed that the degradation of the material after the fabrication processes (heat-treated and coated) does not cause acute toxicity to the local environment of the implant.

To this end, we have developed a biodegradable bone fixation alloy, Mg-1.2Zn-0.5Ca-0.5Mn (wt.%), and post-fabrication processes (i.e., coatings) that showed promising mechanical properties, corrosion behavior, and acceptable in vitro biocompatibility. This work describes the microstructure and corrosion characterization of a novel Mg-Zn-Ca-Mn-based alloy and novel post-fabrication methods (heat treatment and coating), and demonstrates in vivo biocompatibility within a rabbit femoral defect model. Rods representing the post-fabrication stages of our work path for the production of the Mg-1.2Zn-0.5Ca-0.5Mn (wt.%) bone fixation material, i.e., (i) heat-treated alloy rods and (ii) heat-treated and MAO-coated alloy rods, were tested for their in vivo biocompatibility.

## 2. Materials and Methods

### 2.1. Alloy Preparation

The Mg-1.2Zn-0.5Ca-0.5Mn (wt.%) alloy was produced using commercially pure Mg, pure Zn, and a 15% Ca-Mg master alloy, according to the procedure described in [26,27]. The alloy mixing was performed in a steel crucible under a CO_2_ + 0.5% SF6 protective gas atmosphere. The molten mixture was held at 720 °C for 15 min, and cylindrical ingots were then cast into a steel permanent mold at room temperature. Rods of 3 mm diameter were machined out of the ingots for further post-fabrication processes.

### 2.2. Heat Treatment

The alloy rods (Ø3.00 ± 0.05 mm) were solution-treated at 510 °C for 3 h in a tube furnace under vacuum followed by quenching into distilled water at room temperature [28,36]. The samples were subsequently age-hardened in a heat bath of silicone oil at 200 °C for 3 h. The developed thin oxide layer was polished out using SiC of grit size 400–1000 followed by rinsing in an ethanol bath and was then left to dry. The heat-treated rods were then cut down to 6.0 ± 0.1 mm long pins for further processing and testing.

### 2.3. Surface Treatment

The heat-treated samples were coated using the MAO process before cutting using a pulsed DC current of 5000 Hz at a current density of 77 mA/cm^2^ for a total of 5 min. An alkaline phosphate electrolyte solution composed of 3 g/L (NaPO_3_)6 + 8 g/L KF-2H_2_O was used in this study [37,38]. The addition of Fluoride (F) phases is known to increase the conductivity of the electrolyte solution, decrease the breakdown voltage, and extend the window of opportunity for the deposition of MAO coatings with desirable characteristics. The pH of the electrolyte was adjusted to 11 using NaOH solution. The maximum voltage reached was 430 V. The coated rods were then cut into pins of 6.0 ± 0.1 mm in length for the in vivo assessment.

### 2.4. Scanning Electron Microscopy Imaging

The microstructure of the Mg alloy was investigated before and after the post-fabrication methods by using scanning electron microscopy (SEM). The coated samples were tested without preparation, but the alloy and heat-treated samples were mounted in a polymeric material and then polished using 180–2000 grit SiC papers in 90° parallel lines. The samples were then polished using a Leco (St. Joseph, MI, USA) imperial cloth pad with silica powder (0.05 µm) until a mirror surface was obtained. Just before the investigation, samples were etched in acetic glycol solution then quickly rinsed with water and wiped. Finally, all samples were immersed in ethanol and ultrasonically cleaned for 3–5 min before the SEM investigation using a Hitachi (Tokyo, Japan) S-4800.

### 2.5. In Vitro Electrochemical Corrosion Test

The potentiodynamic polarization test (PDP) was used to measure the relative change in the corrosion rate due to the heat treatment and surface coating [16]. Samples of 15 mm diameter and 3 mm thickness of the uncoated samples were prepared and polished using 180–2000 grit SiC paper. A Gamry (Centennial, CO, USA) Instruments Potentiostat was used to perform the test in a simulated body fluid (SBF) at pH 7.4, 37 ± 1 °C and a constant scan rate of 0.5 mV/s. The SBF was prepared according to the procedure in [39]. A standard three-electrode system was used, with saturated silver calomel as the reference electrode, graphite rod as the counter electrode, and the sample as the working electrode [16,37,40,41,42].

### 2.6. Microhardness Test

The microhardness values of the different Mg-Zn-Ca-Mn-based alloy samples (as-cast, heat-treated) were measured using an HMV-G Series Shimadzu microhardness tester (1 kg load cell). Tests were conducted according to the ASTM E384-17 standard. The test employed a 0.5 N load and 15 s as a dwell period [40]. Each sample was tested 10 times before the average value was determined.

### 2.7. In Vivo Biocompatabiltiy Investegation

#### 2.7.1. Animals

Six New Zealand White (NZW) rabbits, female, aged 10–12 weeks and weighing approximately 3 kg, were utilized for this study. Rabbits were housed, handled, and cared for according to institutional guidelines. All surgical procedures were performed in accordance with an approved Institutional Animal Care and Use Committee (IACUC) protocol, animal welfare guidelines (protocol number 2816), and good clinical practice of surgery.

#### 2.7.2. Surgical Procedure

To facilitate surgery, rabbits were pre-medicated with 0.1 mg/kg hydromorphone (Hydromorphone HCl injection 2 mg/mL, Baxter Healthcare Corporation, Deerfield, IL, USA) and 1.0 mg/kg midazolam (Midazolam HCl injection 5 mg/mL, Akorn, Lake Forest, IL, USA) intramuscularly (IM) and provided inhalant oxygen and gas anesthesia (isoflurane) via face mask. Following pre-medication, rabbits had intravenous (IV) catheters placed in a lateral ear vein, were intubated, and had anesthesia maintained with inhalant oxygen and isoflurane, a constant rate infusion (CRI) of 50 µg/kg/minute lidocaine (Lidocaine HCl injection 10 mg/mL, Pfizer, Inc. Lake Forest, IL, USA) IV, and an intra-operative dose of 0.05 mg/kg hydromorphone IV. Experimental metal alloy implants were surgically implanted into NZW rabbits (n = 6) utilizing an established rabbit femoral condyle defect model [43,44,45,46], see Figure 1. The surgical procedure was as follows: a 2.0 cm skin incision was created on the lateral aspect of the left hindlimb, overlying the distal femur, approximately 0.5 cm caudal and parallel to the lateral trochlear ridge of the femur. A 1.0 cm myotomy was performed to expose the bony surface of the lateral femoral condyle. Once exposed, two unicortical 3.0 mm drill holes were created and immediately filled with either a coated or uncoated experimental implant. Each type of implant (coated, uncoated) was placed into each of the defects as determined randomly (1 of each type of implant in every rabbit). Implants were assigned to either the proximal (closer to the diaphysis) or distal (closer to the knee joint), referring to their location within the distal femur. Surgical sites were closed with 5-0 monofilament sutures (PDS, Ethicon Inc. Raritan, NJ, USA) in standard fashion. Also, one coated sample and one uncoated sample were implanted subcutaneously in a few rabbits to provide additional information on the degradation rates of the two tested groups. Immediately post-operatively, 1 mg/kg of meloxicam (Meloxicam 5 mg/mL solution for injection, Boehringer-Ingelheim Vetmedica, St. Joseph, MO, USA) was administered subcutaneously (SQ) and 5 mg/kg of enrofloxacin (Enrofloxacin antibacterial injectable solution 2.27%, Dechra, Overland Park, KS, USA) was administered SQ to prevent post-operative discomfort and infection, respectively.

#### 2.7.3. Post-Operative Care and Management

To maintain adequate comfort, rabbits received 0.1 mg/kg hydromorphone IM every 6 h for the first three days post-operatively and 1 mg/kg meloxicam (Meloxicam 1.5 mg/mL oral suspension, VetOne, Boise, ID, USA) orally (PO) every 24 h for the first seven days post-operatively. To prevent infection, rabbits received 5 mg/kg enrofloxacin (compounded 20.5 mg/mL oral solution, University of Tennessee College of Veterinary Medicine Pharmacy) PO every 12 h for the first three days post-operatively. Rabbits were individually housed with free choice water, hay, and commercial rabbit diet, according to IACUC guidelines. One and two months post-operatively, radiographs of the left hindlimb were obtained to evaluate for the presence of abnormal bony changes and also to evaluate implant appearance and orientation. Rabbits were maintained for either one or two months, (n = 3/timepoint), at which time animals were humanely euthanized.

#### 2.7.4. Focused Post-Mortem Examination

Following euthanasia, post-mortem examinations were performed on the surgical limbs of each animal. During this time, implants were retrieved from the bone. Implants were cleaned of animal tissue if applicable, then were rinsed in ethyl alcohol and saved for material analysis. Femoral segments containing the two surgical defects were harvested and processed for histological examination.

#### 2.7.5. Histological Analysis

Segments of the left distal femur from each animal were fixed in 10% neutral buffered formalin (NBF) for 48 h. Segments were then decalcified in Formical-2000 until they were able to be sliced smoothly with a scalpel blade. After decalcification, bone segments were transferred back to 10% NBF and were submitted for histology (University of Tennessee College of Veterinary Medicine, Veterinary Diagnostic Laboratory, Histopathology Service). Bone segments were embedded within paraffin, and 4 μm-thick decalcified sections were obtained and stained with Hematoxylin and Eosin (H&E). H&E-stained slides were examined microscopically. Samples were evaluated qualitatively by investigators and reviewed by a blinded, board-certified veterinary pathologist. Samples were evaluated for the presence of inflammation, exuberant fibrous connective tissue, and other evidence of foreign body reaction including white blood cell infiltration and the presence of multinucleated giant cells involving and surrounding the defect site. Once characterized, defect sites were compared within and between rabbits to ensure individual animal immune response was appropriately considered.

## 3. Results and Discussion

### 3.1. Microstructure and Surface Characteristics

Figure 2A shows the SEM imaging of the microstructure of the as-cast alloy. A uniform distribution of the Ca_2_Mg_6_Zn_3_ secondary as lamellar eutectoids along the grain boundaries, as well as spherical eutectic colonies throughout the α-Mg matrix phase, can be observed for the as-cast alloy. This suggests that the initial nucleation of Mg dendrites occurs first, and the creation of the eutectic phase is connected to the solidification of the Mg dendritic phase. Figure 2B shows an improvement in the distribution of the Ca_2_Mg_6_Zn_3_ secondary intermetallic phase as well as the uniform grain size in the heat-treated group. The uniform distribution of the Ca_2_Mg_6_Zn_3_ secondary intermetallic phase in the heat-treated group suggests better mechanical and corrosion properties for the heat-treated group. A uniform and finer dispersion of the cathodic phases improves the corrosion behavior by reducing the internal galvanic corrosion effects. A uniform grain size is also beneficial in slowing the corrosion stress cracking propagation. This microstructural analysis on the Mg alloy phases showed a notable improvement and a better degradation behaviour after the heat treatment of the as-cast alloys. Figure 2C,D shows the MAO coating layer. A notable porosity was created by the micro arcs on the outer portion of the coating layer. The average pore size was measured to be 4 ± 2 µm, with the majority of pores (92%) having a size ranging from 1 to 6 µm and the rest of the pores (8%) having a size of 6–11 µm in diameter. The porous nature of the coating layer suggests a good biointegration property, which is beneficial for bone tissue support and cell proliferation. The good biointegration property of the MAO-coated surface was also confirmed by the static water contact angle measurements from our previous work [24]. The results showed a significant enhanced hydrophilicity of the MAO-coated surface (contact angle = 29.6 ± 3.7°) compared to the uncoated surface (contact angle = 102.4 ± 4.4°). This biocompatible coating layer is also hypothesized to provide more of the needed vascularization to the healing zone.

### 3.2. Electrochemical Corrosion and Microhardness Results

Figure 3A shows the in vitro PDP electrochemical corrosion test results of the Mg-1.2Zn-0.5Ca-0.5Mn alloy groups (as-cast, heat-treated, heat-treated and MAO-coated) and Table 1 lists the corrosion characteristics of the PDP electrochemical corrosion test for the different Mg-1.2Zn-0.5Ca-0.5Mn alloy groups.

The heat-treated alloy samples have a slightly lower corrosion current density and showed an increased positive corrosion potential when compared to the as-cast alloy, as shown in Figure 3A. For the heat-treated and MAO-coated alloy sample, a major improvement was noted in both the corrosion current density and corrosion potential, as shown in Figure 3A.

These electrochemical corrosion results reflect the expected reduction in the corrosion rate in vitro and the degradation time of the implanted pins in vivo. Also, through careful examination of the corrosion characteristics of the alloys shown in Table 1, we can see that the current density values went down after the heat treatment process of the as-cast alloy from 290 µA/cm^2^ to 231 µA/cm^2^ and the corrosion potential values increased more positively from −1.83 V to −1.75 V. This confirms that the heat treatment process enhanced the electrochemical corrosion properties and led to a lower corrosion rate (CR) of 10.34 mm/year for the heat-treated alloy sample compared to 12.97 mm/year for the as-cast alloy sample, which can be contributed to the uniform and finer dispersion of the cathodic secondary intermetallic phases in the microstructure, as shown in Figure 2. For instance, the microstructure of the as-cast alloy shows larger regions of the Ca_2_Mg_6_Zn_3_ secondary phase that promotes galvanic corrosion. Meanwhile, the Ca_2_Mg_6_Zn_3_ secondary phase is in the form of finely dispersed precipitates throughout the primary Mg phase of the heat-treated alloy, leading to a reduced galvanic corrosion effect. For the heat-treated and MAO-coated alloy sample, a major improvement was noted in both the corrosion current density and corrosion potential, as shown in Figure 3A. The value of the current density decreased to as low as 0.227 µA/cm^2^ and the corrosion potential value increased positively to -1.60 V. This shows that the combined effect of heat treatment and coating on the as-cast alloy samples improved the corrosion properties significantly. Comparing our results to similar research publications, Bakhsheshi-Rad et al. demonstrated, in their results, the corrosion properties of different Mg alloys without any post-fabrication treatment [47]. For Mg-xZn alloys, the current density started at 370.7 µA/cm^2^ for pure Mg and decreased to 212.4 µA/cm^2^ by increasing the Zn content as in Mg-4Zn. For Mg-Ca-xZn alloys, the current density decreased to 195.5 µA/cm^2^ at zero Zn-content (Mg-0.8Ca) but kept increasing until it reached 317.8 µA/cm^2^ by increasing the Zn content as in Mg-0.8Ca-4Zn. The corrosion rates are directly proportional to the current and follow the same trend. The corrosion rate value was approximately 8.47 mm/yr for pure Mg, reaching its lowest value at 4.12 mm/yr for Mg-0.8Ca-1.25Zn. Dou et al. used MAO coating only to prepare Ca-P and Ca-P-Si coatings on an Mg-1.78Zn-0.51Ca alloy [48]. They managed to decrease the current density from 428.7 µA/cm^2^ for the Mg alloy substrate to 5.251 µA/cm^2^ after the Ca-P coating and 1.322 µA/cm^2^ after the Ca-P-Si coating.

Figure 3B shows the results of the microhardness of the Mg alloys (as-cast and heat-treated) as measured by Shimadzu HMV, which reflect the effect of the heat treatment process on their mechanical properties [40]. The value of the microhardness increased from 71.9 ± 5.4 HV for the as-cast Mg alloy (without any post-fabrication treatment) to as high as 98.1 ± 6.5 HV for the heat-treated Mg alloy. This represents a 36.4% increase in the microhardness of the Mg-1.2Zn-0.5Ca-0.5Mn (wt.%) alloy after the heat treatment process. These values indicate that the developed heat treatment process can be utilized to significantly enhance the surface hardness and wear resistance of Mg alloys.

### 3.3. In Vivo Biocompitability Investigation

#### 3.3.1. Animal Observation and Care

No anesthetic complications occurred, all of the animals recovered well and appeared comfortable. The incisions healed without evidence of surgical site infections and the animals gained weight throughout the study period. One rabbit (#28, timepoint = two months) developed a nonpainful, soft, subcutaneous swelling on the left lateral hindlimb. One rabbit (#22, timepoint = one month) experienced weight loss secondary to reduced appetite towards the end of the study period. Following scheduled humane euthanasia and focused post-mortem examination, this rabbit underwent a full post-mortem exam that revealed a dental malocclusion. This malocclusion was unrelated to the surgical procedure or experimental design and the rabbit received appropriate veterinary care to combat its reduced appetite during the study period.

#### 3.3.2. Radiographic Examination

Radiographs that were obtained one and two months post-operatively (Figure 4) documented the presence and location of the experimental implants located within the distal femurs of the rabbits. There was no radiographic evidence of periosteal reaction, bone destruction, or abnormal bone proliferation noted. For the pins implanted subcutaneously, the faster degradation rate of the uncoated group was clearly observed, as shown in Figure 4D for rabbit #27 after one month of implantation. Also, the local release of hydrogen, as a result of the corrosion process of Mg in aqueous environments, generated gas pockets in the tissue. These pockets are usually observed after implantation and were found to disappear after from three to four months of implantation [49,50]. This emphasizes the impact of the coating on decreasing the degradation rates in order to decrease the release of hydrogen during the healing process.

#### 3.3.3. Post-Mortem Examination Findings

One-month timepoint: The findings at the femoral implantation sites were similar among the rabbits (n = 3). The bone implants were visualized easily and were securely in place. The implants were firmly embedded within the bone and the surrounding soft tissues appeared normal. No bruising, discoloration, or evidence of inflammatory response were detected grossly, as shown in Figure 5. The implants required a mild amount of force to be removed from the bone using forceps. Once removed, it was noted that the distal experimental implant (coated) in rabbit #28 was irregularly corroded. This was most prominent towards the superficial surface of the implant.

Two-month timepoint: The findings at the femoral implantation sites were similar among the rabbits (n = 3). The bone implants were visualized fairly easily and appeared to be well-integrated into the surrounding bone, with the exception of the distal bone implant (uncoated) in rabbit #27, which had a significant capsule of tissue overlying the implant site, and required clearance prior to direct visualization of the indwelling implant. The surrounding tissues in all of animals (n = 3) appeared healthy and within normal limits. The bone implants were significantly more challenging to remove than at the one-month timepoint. Implant removal in these rabbits required the bone to be split and sectioned close to each implant, followed by a firm, constant traction and twisting to completely remove the implants. One animal (rabbit #25) had a boney proliferation along the internal surface of the proximal bone implant. The difference between the uncoated and coated implants was subtle, as the coating was largely no longer present upon gross examination.

#### 3.3.4. Histological Analysis

One-month timepoint: For the uncoated experimental implants in bone within the one-month timepoint, there were mild fibrous connective tissue rims formed surrounding the defect sites, and mild to moderate lymphoplasmacytic cellular infiltration within the connective tissue and surrounding area. These findings were particularly evident within the proximal section of rabbit #28, with that site demonstrating a moderate amount of lymphoplasmacytic cellular infiltration and a moderate amount of fibrous connective tissue formation within the defect site (Figure 6). The coated experimental implants demonstrated minimal fibrous connective tissue formation and minimal lymphoplasmacytic cellular infiltration. No sample demonstrated evidence of infection or necrosis, or evidence of a foreign body reaction. The soft tissue samples were characterized by connective tissue, with minimal lymphoplasmacytic cellular infiltration. No evidence of a foreign body reaction was detected within the soft tissue samples (Figure 7).

Two-month timepoint: Both the coated and uncoated experimental implants in bone within the two-month timepoint demonstrated minimal amounts of fibrous connective tissue and mild evidence of lymphoplasmacytic cellular infiltration. The bone samples did not demonstrate any evidence of infection, necrosis, or a foreign body reaction. A sample of bordering soft tissues from one rabbit (rabbit #27) demonstrated the presence of multinucleated giant cells (Figure 8).

Overall, the post-mortem examinations and histological analysis of experimental implants within the bones of NZW rabbits (n = 6) displayed evidence of mild to moderate inflammatory responses to uncoated implants at the one-month timepoint, no evidence of a foreign body response or inflammatory reaction to coated implants at the one-month timepoint, and no evidence of a foreign body response or inflammatory reactions to coated or uncoated implants at the two-month timepoint. Occasional soft tissue responses to experimental implants were observed; most notably, these responses were comprised of emphysema or air-filled tissue pouches. The histological examination of one such pouch demonstrated the presence of multinucleated giant cells. With the exception of the bone implant site of rabbit #28, which demonstrated the presence of necrotic debris, there was little intra or inter-rabbit variability in the bone implant sites. These findings, along with the findings of occasional irregularly corroded implants, suggest that coating and control of the degradation rate are highly important parameters in the biocompatibility of such Mg-based alloys.

## 4. Conclusions

The basic intent of this article was to produce a strengthened and corrosion-controlled Mg-Zn-Ca-Mn-based alloy and to investigate the effect of the utilized post-fabrication methods (heat treatment and coating) on the in vivo biocompatibility within a rabbit femoral defect model. The mechanical and corrosion properties of the treated alloys are proven to be enhanced by the heat treatment and coating through the results of the microhardness testing compared to the as-cast alloys. The in vivo assessments of the described Mg alloy and fabrication method demonstrate overall appropriate biocompatibility. There was minimal evidence of irregular degradation or undesirable host response to the experimental implants. Where there was evidence of inappropriate host tissue response, there was also evidence of irregularly degraded uncoated implants, suggesting that the coating, purity, and degradation rates of the experimental Mg-based alloys are of critical importance. The results of this study show that it is possible to produce biocompatible Mg-based implants with stronger and more corrosion-controlled properties. These findings are relevant, as they provide a framework to pursue further investigation of coated Mg-based alloys to be utilized in bone stabilization and the regeneration of small extremities. Another long-term objective of this research will be focused on the development of biodegradable and personalized bone fixation implants, leading to a clinical breakthrough. In this approach, patients’ data (e.g., CT and MRI images) can be used to design and fabricate personalized fixation implants based on the strengthened Mg alloy, heat treatment, and coating approaches investigated in this work.

## Figures and Tables

**Figure 1 bioengineering-10-00877-f001:**
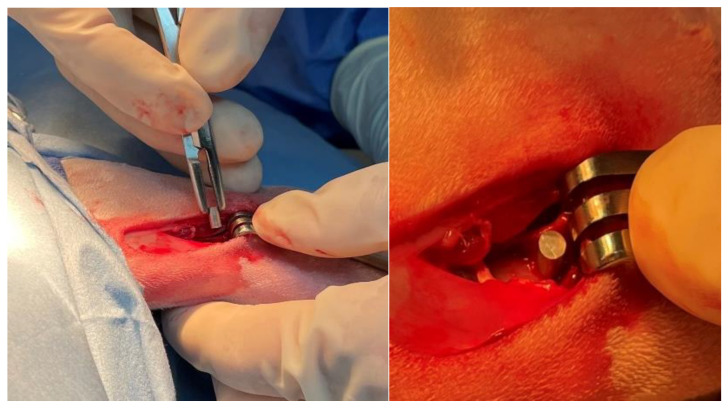
Intra-operative image of the Mg-1.2Zn-0.5Ca alloy implant being placed into lateral femoral condyle of a rabbit.

**Figure 2 bioengineering-10-00877-f002:**
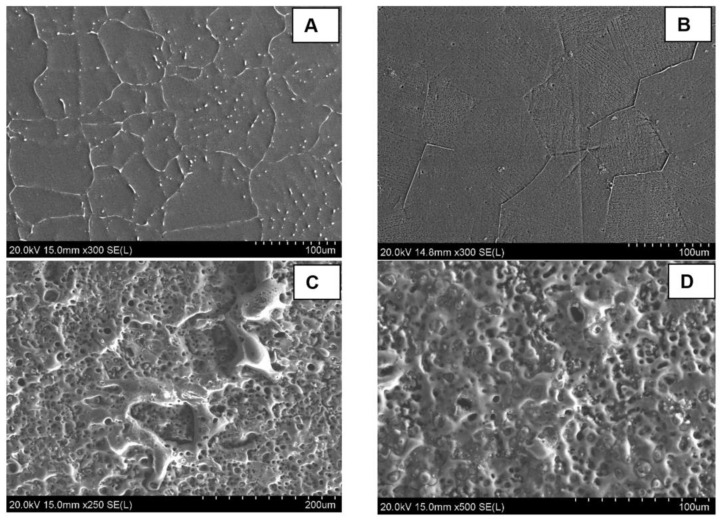
SEM imaging of the surface microstructure for uncoated and MAO-coated samples of Mg-1.2Zn-0.5Ca alloy; (**A**) shows the as-cast microstructure, (**B**) shows the as heat-treated microstructure with a finer dispersion of the secondary intermetallic phases into the grains, and (**C**,**D**) show the porous MAO coating layer on the coated group.

**Figure 3 bioengineering-10-00877-f003:**
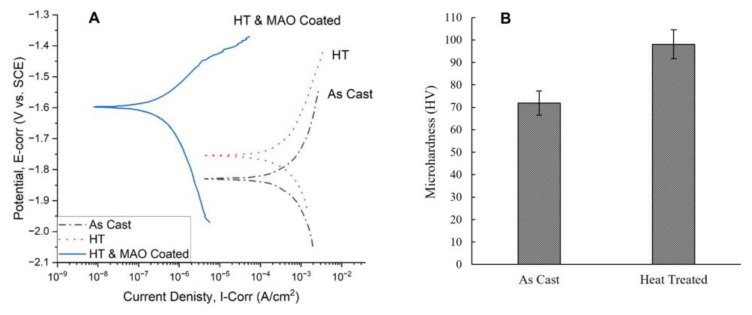
(**A**) PDP Tafel curves comparison between different Mg-1.2Zn-0.5Ca-0.5Mn alloy groups (as-cast, heat-treated, heat-treated and MAO-coated), showing an enhancement in the corrosion characteristics after heat treatment and a significant enhancement after the MAO coating. (**B**) microhardness measurements of different Mg-1.2Zn-0.5Ca-0.5Mn alloy groups (as-cast and heat-treated), showing a significant enhancement in the microhardness after heat treatment.

**Figure 4 bioengineering-10-00877-f004:**
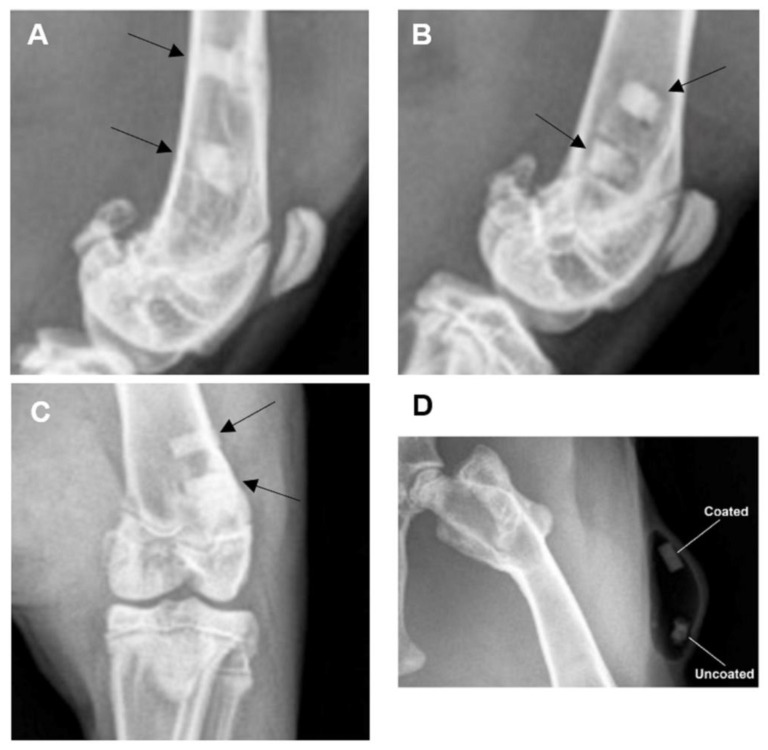
Radiographs demonstrating presence and location of experimental implants. (**A**) Laterally oriented radiographic image (rabbit #24, timepoint = one month). (**B**) Laterally oriented radiographic image (rabbit #25, timepoint = one month). (**C**) Cranial-caudally oriented radiographic image (rabbit #27, timepoint = two months). Black arrows aimed at experimental implants within the bone (distal femur). (**D**) Radiographs demonstrating faster degradation rates for the uncoated implants with pocket resulting from hydrogen evolution (rabbit #27, timepoint = one month).

**Figure 5 bioengineering-10-00877-f005:**
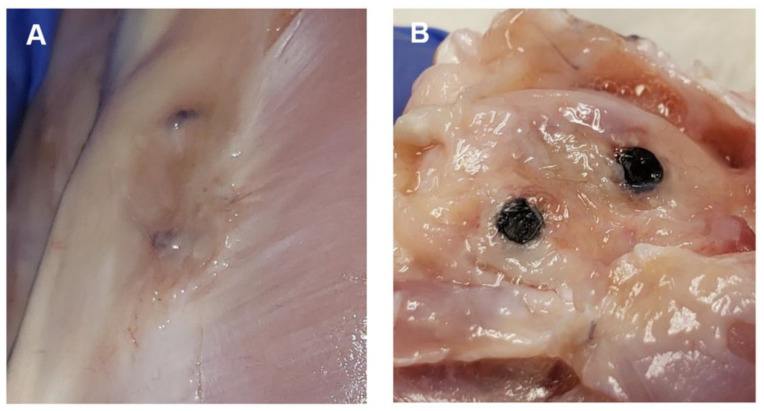
Representative images of post-mortem examinations. (**A**) Image displays the superficial portions of two experimental implants that are within the bone of a partially dissected left hindlimb. Implants are still covered with connective tissue. (**B**) Image displays two experimental implants within bone (distal femur). Connective tissue has been dissected to expose bone and implants.

**Figure 6 bioengineering-10-00877-f006:**
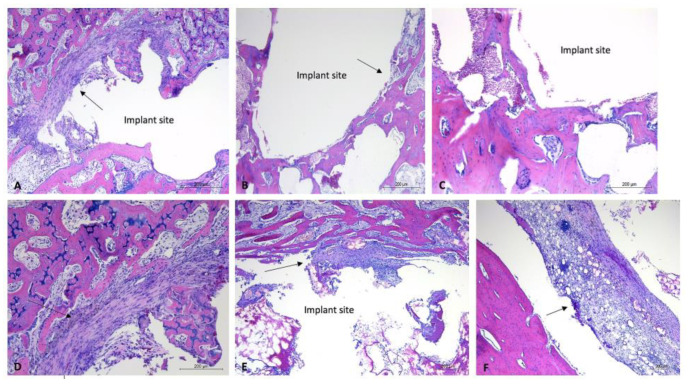
Histology images of bone sections containing uncoated implants, one-month timepoint. H&E-stained sections of bone samples demonstrate defects (with implant removed) visualized as (relatively) circular areas of empty (white) space, surrounded by tissue (bone, connective tissue, adipose). (**A**,**D**): Evidence of fibrous connective tissue formation around edge of implant site with moderate lymphoplasmacytic infiltration. (**A**): 5× magnification, (**D**): 10× magnification. (**B**,**C**): Minimal connective tissue formation within implant site. (**B**): 5× magnification, (**C**): 10× magnification. (**E**,**F**): Evidence of moderate fibrous connective tissue formation and moderate lymphoplasmacytic cellular infiltration. E: 5× magnification, (**F**): 10× magnification. Black arrows aimed at areas of (fibrous) connective tissue. Implant sites labeled “implant site.”

**Figure 7 bioengineering-10-00877-f007:**
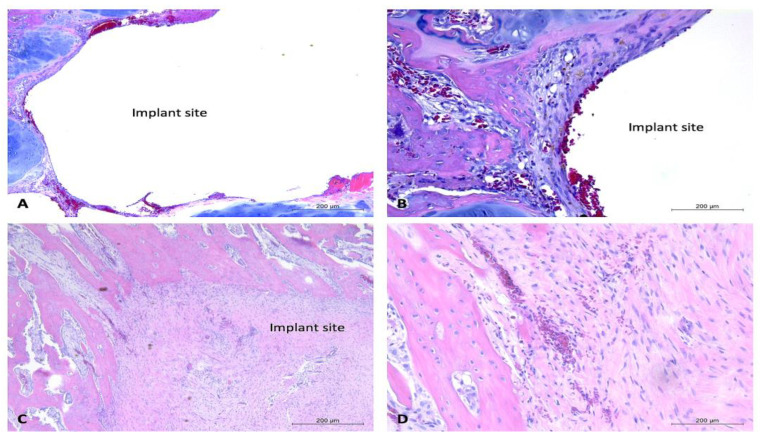
Histology images of bone sections containing coated implants and soft tissue response to uncoated implants, one-month timepoint. H&E-stained sections of bone samples demonstrate defects (with implant removed) visualized as (relatively) circular areas of empty (white) space, surrounded by tissue (bone, connective tissue, adipose, and cartilage). (**A**,**B**): Demonstrate bone defect site with minimal connective tissue formation and mild lymphoplasmacytic cellular infiltration. Vascularization prominent throughout connective tissue. (**A**): 5× magnification, (**B**): 10× magnification. (**C**,**D**): Section containing bone defect site and overlying soft tissues. Demonstrates vascularization throughout soft tissues and presence of healthy connective tissue with minimal evidence of lymphoplasmacytic cellular infiltration. (**C**): 5× magnification, (**D**): 10× magnification. Black arrow aimed at connective tissue. Implant sites labeled “implant site.”

**Figure 8 bioengineering-10-00877-f008:**
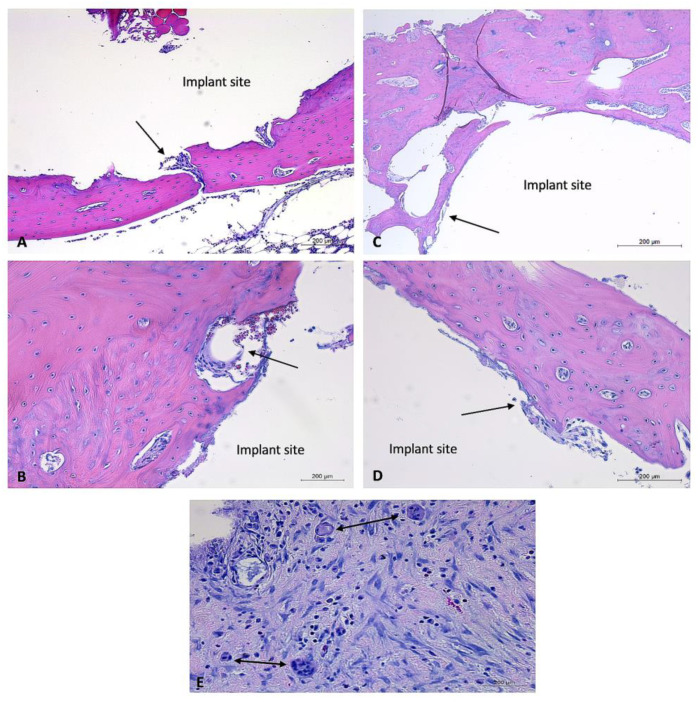
Histology images of bone sections containing coated and uncoated implants and soft tissue response to coated and uncoated implants at two-month timepoint. H&E-stained sections of bone samples demonstrate defects (with implant removed) visualized as (relatively) circular areas of empty (white) space, surrounded by tissue (bone, connective tissue, adipose). (**A**,**B**): Bone defect site that previously contained an uncoated experimental implant. Minimal connective tissue formation and mild lymphoplasmacytic cellular infiltration. Vascularization prominent throughout connective tissue. (**A**): 5× magnification, (**B**): 10× magnification. (**C**,**D**): Bone defect site that previously contained a coated experimental implant. Minimal connective tissue formation and mild lymphoplasmacytic cellular infiltration. (**C**): 5× magnification, (**D**): 10× magnification, (**E**): soft tissue response to uncoated implant. Demonstrates moderate lymphoplasmacytic infiltration and presence of multiple multinucleated giant cells. Black arrows aimed at connective tissue. Implant sites labeled, “implant site.” Double-headed black arrows aimed at multinucleated giant cells.

**Table 1 bioengineering-10-00877-t001:** Corrosion characteristics of PDP electrochemical corrosion test for Mg-1.2Zn-0.5Ca-0.5Mn alloy groups.

Alloy Sample	I_corr_ (µA/cm^2^)	E_corr_ (V)	β Cathode (V/decade)	β Anode (V/decade)	CR (mm/yr)
As-Cast	290	−1.83	−77.95	33.40	12.97
Heat-Treated	231	−1.75	−50.74	53.60	10.34
Heat-Treated & MAO-Coated	0.227	−1.60	−33867.88	37780.99	0.03

## Data Availability

No new data were created or analyzed in this study. Data sharing is not applicable to this article.

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
