# Peer review of "In Vivo Assessment of High-Strength and Corrosion-Controlled Magnesium-Based Bone Implants"

_bioengineering, 2023, doi:10.3390/bioengineering10070877_

Round 1

Reviewer 1 Report

In the paper entitled “In Vivo Assessment of High-Strength and Corrosion-Controlled Magnesium-Based Bone Implants” the authors developed metallic alloys based on magnesium, zinc, calcium and manganese for bone fracture repair applications. The heat treatment of the fabricated biomaterials, followed by their ceramic coating, improved their performance in terms of resistance to corrosion and biocompatibility. An in vivo evaluation of the designed constructs was performed in rabbit models, which provides remarkable value to the work. In general, I found major revisions are needed before considering the paper suitable for its publication in Bioengineering journal. My main concern is related with the used English style, which must be carefully reviewed. The introduction section also needs to be improved, and some experimental procedures and conclusions reached from the obtained results should be explained with more detail. Following I expose some comments and suggestions that could improve the paper and which I would like the authors to address before consider resubmission:

Statements such as “Medical and surgical management of fractures have improved over time” or “Metallic fixation devices are essential in orthopedic surgery” are obvious and can be removed from the introduction section.

In my opinion the first paragraph of the introduction can be shortened to just one or two sentences. Just mention that metallic implants are widely proposed for the treatment of bone fractures, but their current formulations are still far to be optimal.

In line 45 you defined the symbol of magnesium (Mg). But then you didn´t define the other symbols used during the intro (Ca, Mn, Zn). Standardize the criteria in this regard; in my opinion the symbols of the elements are known by the scientific community, it is not necessary to define them.

“Our previous experimental work demonstrated that a significant increase in mechanical strength and corrosion resistance can be achieved after performing a novel heat treatment process on various Mg-Zn-Ca-based alloy systems compared to the pure Mg and as-cast alloys”. The reference to your previous work is ok, but you should briefly discuss the reason of these formulations? Why did you incorporate zinc and calcium? In some formulations you also used manganese. Why?

You provided a reference to support the low toxicity/biocompatibility of Mg. But what about Mn and Zn? You can see and refer these works where the high biocompatibility of these elements (in this case in the form of nanoparticulated systems) was demonstrated: 10.1021/acs.chemmater.9b04848 and 10.1021/acsami.9b20496.

Line 79: MAO abbreviation is used without previous definition.

I suggest summarizing the explanation about the advantages of MAO coating. Instead, you can briefly mention other commonly-used surface modifications of the implants and their main drawnback in comparison with the strategy you propose.

I missed a short mention to other potential therapeutic approaches for the treatment of bone fractures. To give a more global vision of the state of the art in this field to the readers, you can briefly comment the use of tissue engineering strategies for bone repair. In addition, in many cases the biomaterials designed for these bonetissue engineering strategies include metallic elements such as gadolinium (you can see and refer 10.3390/nano13030501) or iron/zinc (10.1002/adfm.202208940 and 10.1021/acsnano.0c08253) in their formulation, which allows combining these strategies with the work that you propose.

In last paragraph of the introduction you mentioned three in a few lines that the post-fabrication process of your materials consists in their heat treatment and surface coating. Keep only one.

I think that the English style in the Introduction should be carefully reviewed. Many sentences are constructed in a form that results confusing.

How did you establish the composition (weigth percentage of each element) in your alloys? Did you measure the presence of different elements using any characterization technique? Or the reported formulation is based in the amount of precursors used in the fabrication?

When you represent an average value with its associated standard deviation, both values must have the same degree of significance. For example, the diameter of the alloy rods cannnot be displayed as (3 ± 0.05) mm. (3.00 ± 0.05) mm is the correct form. The same for the cutting procedure: (6.0 ± 0.1) mm. Apply throughout the manuscript, please.

Use the international system abbreviations for “hours-hrs” and “sec”: h and s.

Typos: in sections 2.7.2 and 2.7.3 the values of the different magnitudes are not separated from de units. E. g. “0.1mg/kg”, “3.0mm”, “5mg/mL”.

For me it is not clear which is the size of the scale bars in SEM images of Figure 2.

Increase resolution Figure 3. Moreover, font size in this figure is too small in my opinion.

The average values of Table 1 should be provided together with their associated standard deviations.

Corrosion rates (CR) of as-cast and heat-treated alloys showed in Table 1 are different from those displayed in the text (12.97 vs 12.98 and 10.34 vs 10.35, respectively). Although irrelevant differences, it should be corrected.

What about the corrosion rate of MAO-coated alloys?

Section 3.3.1: “This malocclusion was unrelated to surgical procedure or experimental design”. How did you reach this conclusion? A more detailed explanation and/or references should be provided at this point.

Line 342: “Bone implants were significantly more challenging to remove than the one-month timepoint. Implant removal in these rabbits required the bone to be split and sectioned close to each implant, followed by a firm, constant traction and twisting to completely remove the implants.” Can this procedure affect the integrity of the implants? Can you provide images to demonstrate in a qualitative form that the implants removed after two months were not “physically modified” by the aggressive procedure used to extract them?

Scales bars shoul be incorporated in all the histology images.

Line 354, typo: “Hostological”

I suggest incorporating in conclusions section a very short discussion (2-3 sentences) about potential further developments in the field of bone repair based on your strategy. Here you can mention the potential combination of your fabrication approach with 3D printing to design high-resolution and personalized implants (see 10.1039/D1TB00717C), or their modification with materials which allow their tracking through different bioimaging techniques such as fluorescence, MRI, etc.

The English style must be professionally checked. I found several confusing and not well-constructed sentences.

Since I am not a native English speaker, if the editor or other reviewers disagree with this observation, I apologize to the authors.

Reviewer 2 Report

In this study, authors have developed a Mg-Zn-Ca-Mn-based implant to access in vitro corrosion rate as well as in vivo biocompatibility in rabbit model. The effect of post-fabrication processes upon heat treatment and coating was investigated. The implant was characterized in terms of Microhardness test, In vitro electrochemical corrosion test, Scanning electron microscopy imaging before biocompatibility investigation.

However, authors must be addressed the issues:

·        Authors must enrich the abstract with more quantitative results.

·        Write the expanded form of MAO abbreviation at the starting.

·        Authors must present ethical permission/code from authorized committee for Animal study.

·        The mean size of pore after MAO coating should be calculated and mentioned.

·        Authors should present the P value and standard deviation for comparison (Table 1).

·        Authors should present the corrosion mechanism of the coated and bare implant in a solution thermodynamically.

·        As the surface wettability is a key property and determined from the contact angle (CA) between the solid surface and liquid, it should be presented.

·        How authors have characterized the successful MAO coating? By which test?

Best

Moderate editing of English language required

Reviewer 3 Report

1. Can the author test the changes in immune-related indicators in the blood of rabbits after implantation, in addition to immunohistochemistry?

2. The author only used female rabbits in the study, and whether the animal gender would have an impact on the results was considered.

3. The author's explanation of Fig.2 is somewhat simple and requires a deeper discussion.

Round 2

Reviewer 1 Report

My comments were well-addressed by the authors, therefore I recommend the publication of the article in the present form. Just a minor correction: use the international system abbreviations for “hours-hrs” and “sec”: h and s (you didn´t change the abbreviaions in all cases in the corrected version)

Reviewer 2 Report

.

.

Reviewer 3 Report

It can be accepted.